# The Quantification of Paclitaxel and Its Two Major Metabolites in Biological Samples by HPLC-MS/MS and Its Application in a Pharmacokinetic and Tumor Distribution Study in Xenograft Nude Mouse

**DOI:** 10.3390/molecules28031027

**Published:** 2023-01-19

**Authors:** Haijin Huang, Haolin Huang, Yongbing Sun, Qian Liu

**Affiliations:** 1Suzhou Medical College, Soochow University, 199 Ren’ai Road, Suzhou Industrial Park, Suzhou 215123, China; 2Department of Pediatric Surgery, The First Affiliated Hospital of Gannan Medical University, 23 Qingnian Road, Ganzhou 341001, China; 3School of Stomatology, Fuzhou Medical University, 9 Donglin Road, Fuzhou 344000, China; 4Division of Pharmaceutics, Jiangxi University of Traditional Chinese Medicine, 1688 Meiling Road, Nanchang 330004, China; 5School of Life Science, Jiangxi University of Traditional Chinese Medicine, 1688 Meiling Road, Nanchang 330004, China

**Keywords:** Paclitaxel, 6α-hydroxypaclitaxel, *p*-3′-hydroxypaclitaxel, solid-phase extraction, HPLC–MS/MS, pharmacokinetics and tissue distribution

## Abstract

A rapid and sensitive high-performance liquid chromatography–tandem mass spectrometry (HPLC–MS/MS) method was developed for the quantification of Paclitaxel (PTX), 6α-hydroxypaclitaxel (6α-OHP), and *p*-3′-hydroxypaclitaxel (3′-OHP) in mouse plasma and tumor tissue. The analytes were separated using a C18 column (50 × 2.1 mm, 1.8 μm), and a triple-quadrupole mass spectrometry device equipped with an electrospray ionization (ESI) source was applied for their detection. PTX, 6α-OHP, and 3′-OHP were extracted from the biological samples with the solid-phase extraction cartridge. The method was fully validated according to the FDA’s guidance. The method was linear over the concentration ranges of 0.5~1000.0 ng/mL for PTX and 0.25~500.0 ng/mL for 6α-OHP and 3′-OHP. The precision, accuracy, extraction recovery, and matrix effects were within acceptable limits. The present method was successfully applied to the study of the pharmacokinetics and distribution of PTX, 6α-OHP, and 3′-OHP in the tumors of post xenograft nude mice intravenously injected with PTX solution.

## 1. Introduction

Paclitaxel (PTX) is a natural antineoplastic agent that was derived from the bark of Pacific Yew (*Taxus brevifolia*) in the early 1970s, and it was approved by the FDA in 1992. As a very famous drug throughout the world, PTX has been widely applied to the treatment of many cancers such as ovarian cancer, breast cancer, pancreatic cancer, lung cancer, and so on [1,2,3]. PTX can block the mitosis and induce the apoptosis of malignant tumor cells by binding to the *β* subunit of tubulin in the cell and promoting microtubule polymerization and stabilization.

PTX is metabolized in liver by cytochrome P450 (CYP) enzymes. CYP2C8 metabolizes PTX to 6α-hydroxypaclitaxel (6α-OHP) and CYP3A4 to *p*-3′-hydroxypaclitaxel (3′-OHP) [4,5]. PTX treatment is often associated with toxicities such as myelosuppression and neurotoxicity, exhibiting a marked intersubject variablity. A lot of studies have found that the severity of PTX-induced neurotoxicity may be related to its in vivo disposition and its metabolites. [6,7] The cytotoxicity of PTX in HL60 and K562 human leukemia cells was increased in the presence of a noncytotoxic concentration of 6α-OHP. It was also reported that 6α-OHP possessed bone marrow toxicity when tested on human bone marrow cells [8,9]. Therefore, the P450-mediated metabolism of PTX and its possible metabolites are the important factors that affect the clinical efficacy and toxicity of PTX.

Phenolic substances, such as resveratrol, gallic acid, and some flavonoids, are very rich in many fruits and vegetables. As a typical phenolic substance, resveratrol is a very common constituent in grapes, peanuts, and polygonum cuspidatum. Resveratrol can inhibit the metabolic activity of CYP2C8 and CYP3A4 [10]. In many cases, cancer patients may co-administer PTX along with these fruits, which may change the disposition of PTX in vivo. Hence, it is very meaningful to investigate whether resveratrol interferes with the metabolism of PTX, and it is very important to predict potential resveratrol–PTX interactions and the potential induced toxicity.

P450-mediated PTX–resveratrol interactions based on changes in the area under the plasma concentration-time curve (AUC)and other pharmacokinetic (PK) parameters have not been reported [11,12,13]. So, it is indispensable to develop a facile and rapid high-performance liquid chromatography–tandem mass spectrometry (HPLC–MS/MS) method to quantify PTX, 6α-OHP, and 3′-OHP in biological samples to support the PK study and tissue distribution after PTX is administered to mice in the presence or absence of resveratrol.

At present, although there have been many HPLC–MS/MS methods developed to determine the concentration of PTX, 6α-OHP, and 3′-OHP in plasma, these methods have some different disadvantages [14,15,16,17,18,19,20]. The major disadvantage is that the analysis time per run is too long. Hiroaki Yamaguchi developed a column-switching liquid chromatography/tandem mass spectrometry method for the simultaneous determination of PTX, 6α-OHP, and 3′-OHP in plasma [14]. However, the analysis time per run was as long as 25 min. In the present study, we developed a very rapid HPLC–MS/MS method to determine the concentration of PTX, 6α-OHP, and 3′-OHP in rat plasma and tumor tissue homogenates. A CN 96-well solid-phase extraction (SPE) cartridge plate was used to pretreat the biological samples, which was very convenient and could provide very clean samples when compared with liquid–liquid extraction and protein precipitation. The analysis time per run was only 3.0 min, which could satisfy the needs of the high-throughput quantitation of a large number of biological samples. Finally, the present method was successfully applied to the PK study and distribution of PTX, 6α-OHP, and 3′-OHP in the tumors of post xenograft nude mice intravenously injected with PTX solution in the presence of resveratrol.

## 2. Results and Discussion

### 2.1. Method Development

To achieve the high-throughput quantification of a large number of biological samples in an HPLC–MS/MS system, the analysis time for every sample should be as short as possible. So, the samples must be cleanly pretreated to avoid the matrix effect resulting from the co-eluting of polar endogenous components. In the present study, we attempted to pretreat the plasma and the tumor homogenates with different methods, such as solid-phase extraction, liquid–liquid extraction (methyl tert-butyl ether as the extraction solvent), protein precipitation (methanol as the precipitation solvent) and so on. The results showed that the solid-phase extraction could provide a higher extraction recovery rate and a lower ion suppression when compared with the other two pretreatment methods (unpublished data), and thus this method decreased the matrix effect. Therefore, we could decrease the retention time (RT) of the PTX, 6α-OHP, and 3′-OHP in the present HPLC–MS/MS system, and the analysis time per run was only 3.0 min (Table 1), which was shorter than all the methods reported in other publications. Furthermore, the 96-well cartridge plate could realize the batch processing of the samples with a high efficiency. The addition of formic acid in the mobile phase could increase the mass response of the analytes (Figure 1).

**Table 1 molecules-28-01027-t001:** The analysis time per run comparison between the present study and the reported publications.

Analysis Time	Reference
3.0 min	The present study
3.5 min	J. Chromatogr. B, 2003, 785: 253–261. [15]
4.5 min	Anal. Chem. 2005, 77: 4677–4683. [16]
9.0 min	Rapid Commun. Mass Spectrom. 2006, 20: 2183–2189 [17]
9.0 min	Biomed. Chromatogr. 2006, 20: 139–148. [18]
7.0 min	J. Chromatogr. B, 2011, 879: 2018–2022. [19]
25.0 min	Biomed. Chromatogr. 2013, 27: 539–544 [14]
11.5 min	J. Pharm. Biomed. Anal. 2014, 91: 131–137 [20]

### 2.2. Method Validation

#### 2.2.1. Selectivity

The representative HPLC–MS/MS chromatograms of a blank, a spiked biological sample with PTX (0.5 ng/mL), 6α-OHP (0.25 ng/mL), 3′-OHP (0.25 ng/mL) and DTX, and a biological sample from a mouse after an intravenous injection of PTX solution are shown in Figure 2 and Figure 3. No significant interference from the endogenous component in the plasma or tumor homogenate was observed with the PTX, 6α-OHP, 3′-OHP, and DTX, showing that the present method was specific.

#### 2.2.2. Linearity and Lowest Limit of Quantification

The linearity was studied over the concentration range of 0.5~1000.0 ng/mL for PTX and 0.25~500.0 ng/mL for 6α-OHP and 3′-OHP in the mouse plasma and tumor tissue homogenate. The typical equations of the calibration curves are listed in Table 2. All the correlation coefficients (*r*) exceeded 0.99, showing a good linearity over the concentration range (Table 2). The LLOQ was 0.5 ng/mL for PTX and 0.25 ng/mL for 6α-OHP and 3′-OHP. The precision and accuracy at the LLOQ level were within the accepted limits.

#### 2.2.3. Precision and Accuracy

Table 3 summarizes the intra- and inter-day precision and accuracy for PTX, 6α-OHP, and 3′-OHP in the QC samples. The intra- and inter-day RSD values were below 11.1%, and the accuracy ranged from −8.0% to 11.0%. All the results met the FDA criteria for a bioanalytical method, which showed that the present method was precise and accurate.

#### 2.2.4. Matrix Effect and Extraction Recovery

The mean matrix effects of PTX, 6α-OHP, and 3′-OHP were 97.1 ± 4.3%, 95.3 ± 5.1%, and 104.2 ± 5.9%, respectively. The mean extraction recovery rates of PTX, 6α-OHP, and 3′-OHP were 92.5 ± 4.5%, 95.2 ± 5.5%, and 89.6 ± 5.2% at the QC concentrations levels, respectively. The matrix effect and extraction recovery for IS were 93.4 ± 3.9% and 89.4 ± 4.3%, respectively. From the above results, it can be seen that the solid-phase extraction could provide a clean sample and a high extraction recovery.

#### 2.2.5. Stability

The stability of PTX, 6α-OHP, and 3′-OHP in the mouse plasma and tumor homogenate was evaluated at three QC levels under various conditions. The results showed that PTX, 6α-OHP, and 3′-OHP were stable at 23 °C for 2 h after three complete freeze/thaw cycles (−20 °C to 23 °C) and after long-term sample storage (−20 °C for 30 days). The extracted samples on the autosampler rack at 4 °C were also stable for 24 h.

### 2.3. Application of PK and Tumor Tissue Distribution Study in Xenograft Nude Mice

This validated method was successfully applied to the study of the PK and tumor tissue distribution in post xenograft nude mice injected with PTX solution. The plasma concentration–time profiles of PTX, 6α-OHP, and 3′-OHP are shown in Figure 4 after the injection of PTX solution in the presence or absence of resveratrol, and the PK parameters are provided in Table 4 and Table 5. The oral administration of resveratrol for five days significantly (*p* < 0.05) increased the AUC of PTX by 30.1% and decreased the total plasma clearance (Cl_t_) of PTX by 27.7%. Other parameters were not significantly altered, such as the elimination rate constant (*k*) and the terminal half-life (*t*_1/2_). Accordingly, the AUC of 6α-OHP and 3′-OHP in the presence of resveratrol was significantly lower when compared with AUC in the absence of resveratrol. Therefore, resveratrol could inhibit the metabolism of PTX and decreased the formation of 6α-OHP and 3′-OHP. Furthermore, the concentration of PTX, 6α-OHP, and 3′-OHP in the tumor tissue displayed a similar tendency to that of PTX, 6α-OHP, and 3′-OHP in the plasma (Figure 5) [12,13].

PTX metabolism is catalyzed mainly by CYP3A4 and 2C8. Resveratrol is a substrate of CYP3A4 and 2C8. Since resveratrol has been found in many fruits and foods, we should pay attention to the resveratrol–PTX interaction when cancer patients co-administer PTX along with these fruits or foods.

## 3. Materials and Methods

### 3.1. Drugs and Reagents

Paclitaxel (purity > 99.5%, PTX) was purchased from Shanghai Acmec Biochemical Co., Ltd. (Shanghai, China). Docetaxel (DTX, internal standard, IS, purity > 99.5%) was provided by Wuhan Jingcan Biotechnology Co., Ltd. (Wuhan, China). Resveratrol, 6α-hydroxypaclitaxel (6α-OHP), and *p*-3′-hydroxypaclitaxel (3′-OHP) were purchased from Shanghai Haohong Scientific Co., Ltd. (Shanghai, China). A Phenomenex CN 96-well solid phase extraction cartridge plate was purchased from Tianjin Agela Technologies Co., Ltd. (Tianjin, China). Methanol and acetonitrile were purchased from Fisher Scientific (Pittsburgh, PA, USA). Injectable PTX solution was purchased from Jiangxi Cancer Hospital (Nanchang, Jiangxi). Physiological saline (0.9% sodium chloride solution) was purchased from the affiliated hospital of Jiangxi University of Chinese Medicine (Nanchang, Jiangxi). Formic acid (FA) was purchased from Sigma-Aldrich (St. Louis, MO, USA). HPLC-grade water was prepared using a Mill-Q Simplicity purification system (Millipore Corp., Burlington, MA, USA). All other reagents were of analytical grade.

### 3.2. HPLC-MS/MS Conditions

A HPLC system (Shimadzu corporation, Kyoto, Japan) was used that was composed of an LC-30AD pump and a SIL-30AC autosampler. Chromatographic separation was conducted using a UHPLC XB-C18 Column (2.1 × 50 mm, 1.8 μm, Welch corporation, Shanghai, China), and the column oven was maintained at 30 °C. The mobile phases consisted of water containing 0.1% formic acid (A) and methanol containing 0.1% formic acid (B). The isocratic elution program was in the proportions of A (20%) and B (80%). The flow rate was set at 0.2 mL/min, and the total analysis time per injection was 3.0 min.

The MS analysis was performed on a Triple Quad 5500 system from Applied Biosystems (MDS-Sciex, Concord, ON, Canada) equipped with Turbo V sources and a Turbo Ionspray^TM^ interface. The electrospray ionization (ESI) source was performed in the positive mode, and the mass spectrometric parameters were optimized as follows: Turbo ion spray (TIS) temperature, 500 °C; ion spray voltage, 5500 V; curtain gas: nitrogen at 35 L/min; nebulizing gas, 40 L/min; TIS gas, 40 L/min; entrance potential, 10 V; and collision cell exit potential, 14 V. The declustering potentials (DP) for PTX, 6α-OHP, 3′-OHP, and DTX were adjusted to 110 V, 123 V, 131V, and 60 V, respectively. The optimized collision energies for PTX, 6α-OHP, 3′-OHP, and DTX were adjusted to 24 eV, 23 eV, 23 eV, and 20 eV, respectively. Quantification was performed using multiple reaction monitoring (MRM), and the optimized MRM transitions were 854.4→286.2 for PTX, 870.0→286.0 for 6α-OHP, 870.0→302.1 for 3′-OHP, and 808.5→527.0 for DTX. The quadrupoles Q1 and Q3 were set on the unit resolution. The Analyst Software^TM^ (version 1.6.2) was used to process the obtained data.

### 3.3. Preparation of Standard and Quality Control (QC) Samples

About 0.1 g of tumor tissue was accurately weighed and placed into a 5 mL PE tube. Then, 0.9 mL physiological saline and 0.1 mL methanol were added to the tube. The whole sample was homogenized for 5 min, and the accurate final volume of the mixture was recorded.

Primary stock solutions of PTX (2.0 mg/mL), 6α-OHP (1.0 mg/mL), and 3′-OHP (1.0 mg/mL) were prepared by dissolving the accurately weighed amounts in methanol. DTX stock solution (1.0 mg/mL, internal standard) was also prepared with methanol. Working solutions of PTX, 6α-OHP, and 3′-OHP were prepared by serially diluting the stock solution in methanol. The internal standard solution was diluted with methanol to 100.0 ng/mL.

A 50 μL working solution of PTX, 6α-OHP, and 3′-OHP was added to a 950 μL blank biological sample (blank rat plasma or tumor homogenate) to yield spiked calibration standards. The calibration curves were linear over the concentration range of 0.5~1000.0 ng/mL for PTX and 0.25~500.0 ng/mL for 6α-OHP and 3′-OHP. A 50 μL of spiked calibration standard was used to construct the calibration curve. The nominal concentrations of the quality control (QC) samples at the low, medium, and high level were 1.0, 500.0, and 800.0 ng/mL for PTX and 0.5, 250.0, and 400.0 ng/mL for 6α-OHP and 3′-OHP, respectively, and the QC samples were prepared as the calibration standards.

### 3.4. Sample Preparation

A total of 50 μL of IS solution (100.0 ng/mL) was added to 50 μL aliquots of biological samples (plasma or tumor tissue homogenate). The samples were briefly mixed for 2 min following the addition of 100 μL of ammonium acetate solution (0.2 mM). Then, the samples were transferred to a CN 96-well SPE cartridge plate that was pre-conditioned and equilibrated by 400 μL of methanol followed by 400 μL of ammonium acetate solution (10 mM). The sample was washed with 400 μL of ammonium acetate solution (10 mM) and 400 μL of methanol/10 mM ammonium acetate solution (20:80). Finally, the samples were eluted with 400 μL of acetonitrile, and the eluates were evaporated by N_2_. The residue was reconstituted in a 200 μL mobile phase by vortex mixing for 2 min, and the resulting solution was transferred to an autosampler vial at 4 °C and injected (5 μL) into the HPLC/MS/MS system for the quantitative analysis.

### 3.5. Method Validation

Validation of the method in terms of its selectivity, linearity, precision and accuracy, extraction recovery, matrix effect, and stability was conducted in accordance with the Guidance for Industry, Bioanalytical, Method Validation, US Food and Drug Administration [21].

The specificity was evaluated by analyzing a drug-free biological sample from the six different sources with the corresponding spiked biological samples at the lowest limit of quantification (LLOQ) level as well as samples obtained from the mice after injection with PTX solution. The linearity was determined by analyzing the standard curve samples, which were prepared by spiking PTX, 6α-OHP, and 3′-OHP working solutions into blank biological samples. Precision and accuracy were determined by assessing six replicates of QC samples at three levels on three consecutive days. Precision was expressed as the relative standard deviation (RSD), and accuracy was reported as the relative error (RE). The intra- and inter-day precision values were required to not exceed 15%, and the accuracy was required to be within ± 15%. The extraction recovery and the matrix effect were calculated by analyzing three replicates of spiked biological samples at the same concentrations as the QC samples. The matrix effect was assessed as follows: the analytes were added at three QC concentration levels to the blank matrix from three different individuals, and the peak area (A_m_) was determined and recorded. The peak area (A_s_) of the standard solution at the same concentration was also determined and recorded. The matrix effect (ME) = A_m_/A_s_ × 100%. The stability of PTX, 6α-OHP, and 3′-OHP was assessed at three QC concentration levels (n = 3) under the following conditions: three complete freeze/thaw cycles (−20 to 23 °C), long-term sample storage (−20 °C for 30 days), and bench-top storage (23 °C for 2 h). The ready-to-inject stability of the extracted samples in the autosampler rack at 4 °C for 24 h was also evaluated.

### 3.6. Pharmacokinetics and Tumor Tissue Distribution

Nude mice weighing from 18 to 22 g were used for the PK and tumor tissue distribution study. All animal experiments were performed in accordance with institutional guidelines and were approved by the University Committee on Use and Care of Animals, Jiangxi University of Chinese Medicine. Animals were housed under standard conditions of temperature (25 ± 2 °C), humidity (65 ± 10%), and light. The mice were allowed free movement and had access to food and water for 7 days before the experiments. Mice were fasted overnight with free access to water before the day of the experiment.

A549 cells (5~6 million cells per mouse) were injected subcutaneously close to the lower mammary gland on the right side of the nude mice. The tumors were allowed to grow to about 300 mm^3^. Then, the animals were divided into the control and the treatment groups, wherein the control groups were allowed free access to sucrose-water for drinking and the treatment group was orally administered resveratrol (5 mg/kg) for five consecutive days.

The mice in the control group and in the treatment group were injected with PTX solution (5 mg/kg calculated as PTX). Plasma and tumor tissues were collected at 0, 0.25 h, 0.5 h, 1 h, 2 h, 4 h, 6 h, 8 h, 12 h, and 24 h. At every sampling time point, 5 to 6 nude mice were anesthetized with ether, and blood was collected via cardiac puncture from every mouse. Then, these mice were sacrificed, and all tumors were also collected and homogenized as described in Section 3.3. After centrifugation at 5000 rpm for 10 min, about 100 μL of plasma from every mouse was collected and frozen at −20 °C. The concentrations of PTX, 6α-OHP, and 3′-OHP in the plasma and tumors were determined by HPLC–MS/MS. If the concentration was over the range of the calibration curves, the sample was diluted with blank plasma or tumor homogenate.

### 3.7. Data Analysis

All data were processed by noncompartmental analysis using the DAS 2.0 software package (Chinese Pharmacological Society). The plasma concentration at different times was expressed as the mean ± standard deviation (S.D.), and the mean concentration–time curves were plotted. The maximum plasma concentration (*C*_max_) and *T*_max_ were observed directly from the concentration–time curves. The area under the plasma concentration–time curve from zero to infinity (AUC_0−∞_) was calculated using the linear–trapezoidal rule with extrapolation to infinity.

## 4. Conclusions

A rapid and sensitive HPLC–MS/MS method for the simultaneous determination of PTX, 6α-OHP, and 3′-OHP in mouse plasma and tumor tissue was developed. The 96-well SPE cartridge plate was convenient and could provide very clean samples. The method was applied to the study of the PKs and the tumor distribution in post xenograft nude mice injected with PTX solution in the presence and absence of resveratrol. The present study reported the PTX–phenolic substances interactions based on the AUC change. The oral administration of resveratrol for five days could significantly increase the AUC of PTX and decreased the total plasma clearance of PTX. Resveratrol could also decrease the AUC of 6α-OHP and 3′-OHP.

## Figures and Tables

**Figure 1 molecules-28-01027-f001:**
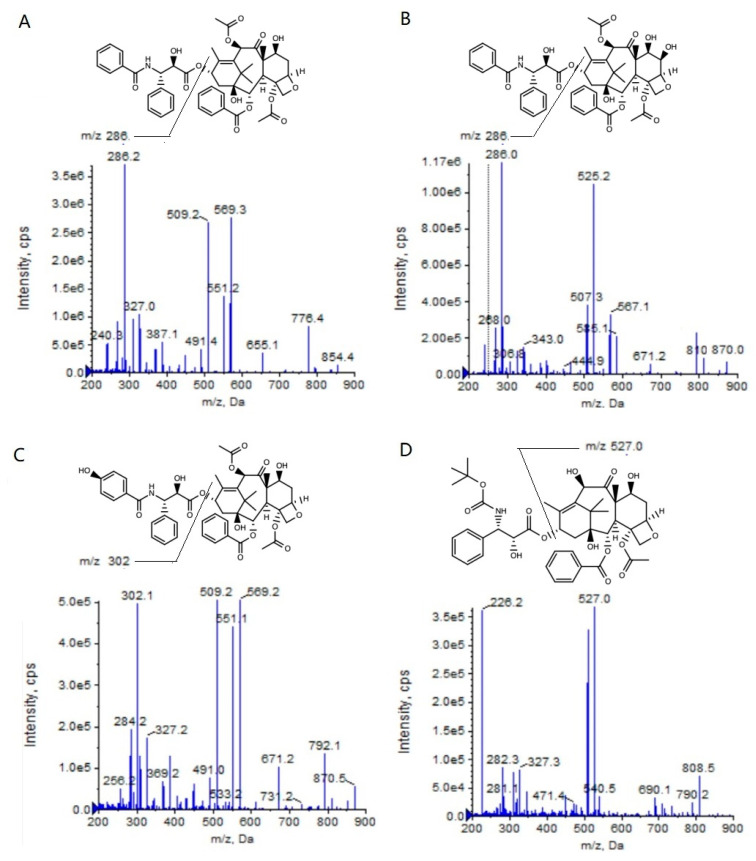
Product ion mass spectra of PTX (**A**), 6α-OHP (**B**), 3′-OHP (**C**), and DTX (**D**).

**Figure 2 molecules-28-01027-f002:**
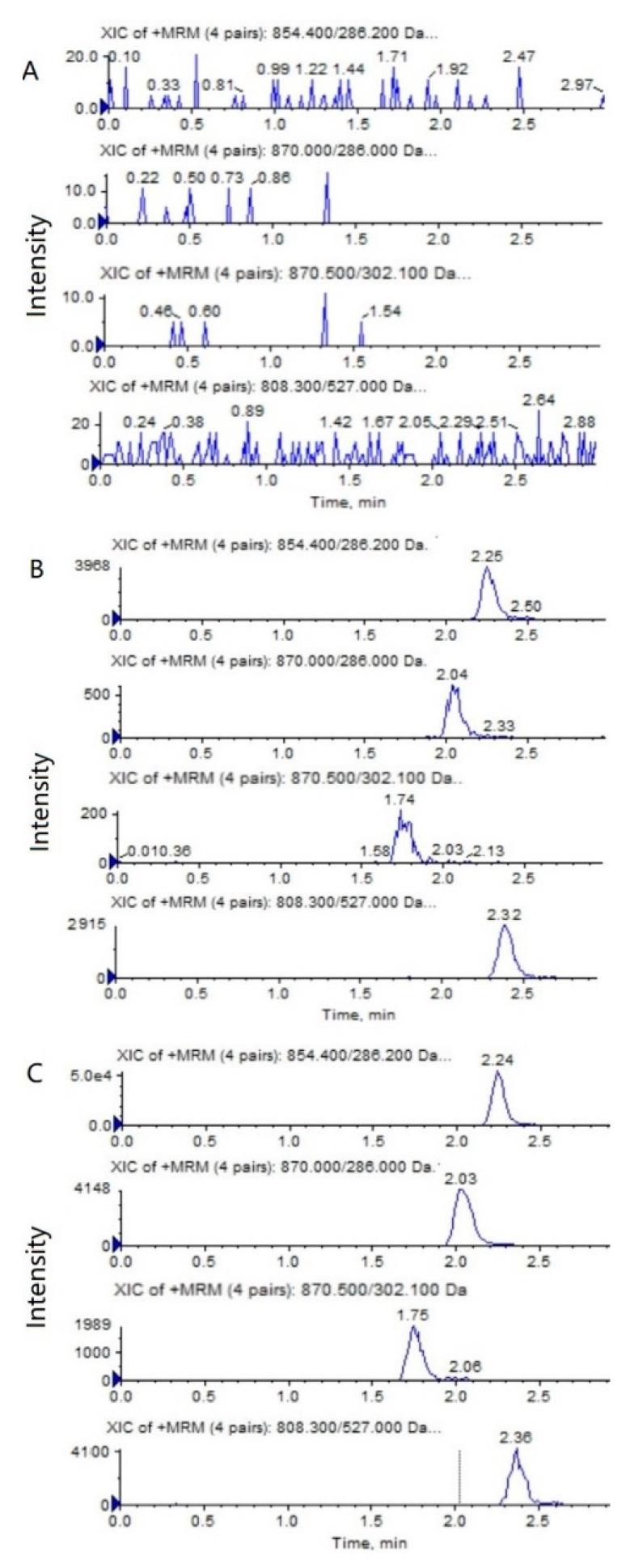
Representative MRM chromatograms of PTX (*m*/*z* 854.4→286.2), 6α-OHP (*m*/*z* 870.0→286.0), 3′-OHP (*m*/*z* 870.0→302.1), and DTX (*m*/*z* 808.5→527.0) in mouse plasma: (**A**) a blank mouse plasma sample; (**B**) plasma samples spiked with PTX (0.5 ng/mL), 6α-OHP (0.25 ng/mL), 3′-OHP (0.25 ng/mL), and the internal standard; (**C**) a mouse plasma sample following the intravenous injection of PTX solution at 5 mg/kg.

**Figure 3 molecules-28-01027-f003:**
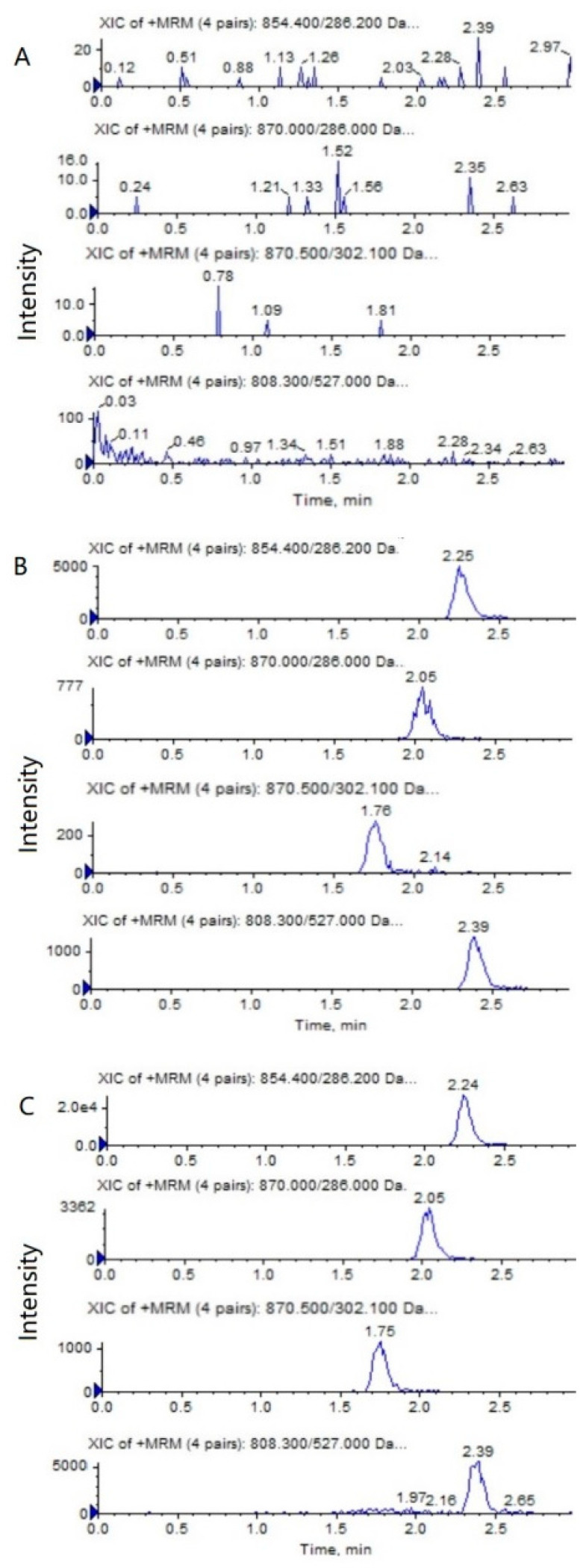
Representative MRM chromatograms of PTX (*m*/*z* 854.4→286.2), 6α-OHP (*m*/*z* 870.0→286.0), 3′-OHP (*m*/*z* 870.0→302.1), and DTX (*m*/*z* 808.5→527.0) in tumor tissue homogenate: (**A**) a blank mouse tumor sample; (**B**) tumor samples spiked with PTX (0.5 ng/mL), 6α-OHP (0.25 ng/mL), 3′-OHP (0.25 ng/mL), and the internal standard; (**C**) a mouse tumor sample following the intravenous injection of PTX solution at 5 mg/kg.

**Figure 4 molecules-28-01027-f004:**
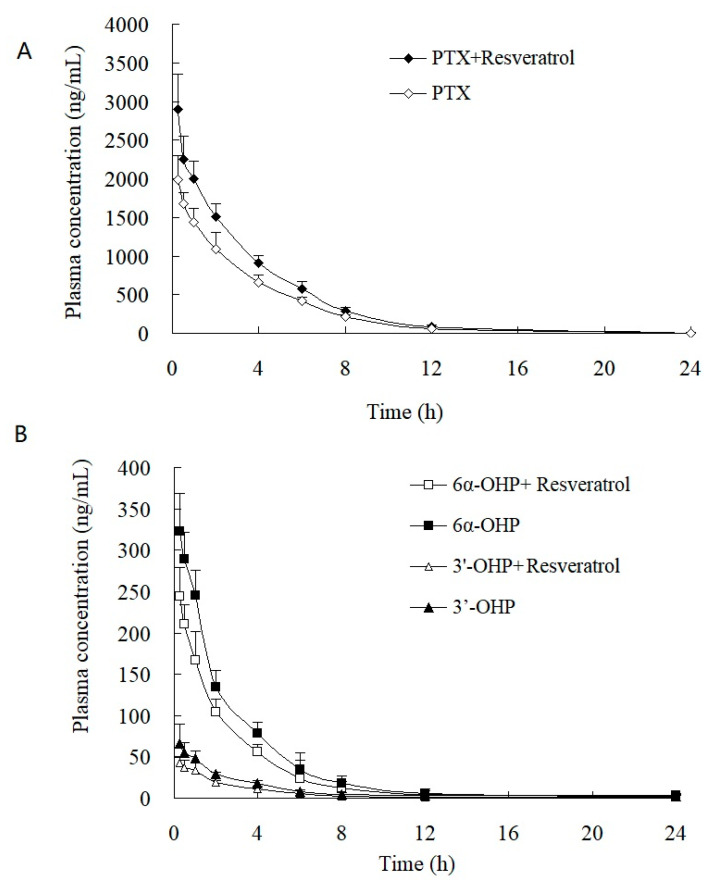
Plasma concentration–time profiles of (**A**) PTX, (**B**) 6α-OHP, and 3′-OHP in post xenograft nude mice intravenously administered with PTX solution at 5 mg/kg. ◆, ☐, and △: the concentration of PTX, 6α-OHP, and 3′-OHP in plasma in the presence of resveratrol, respectively; ◇, ■, and ▲: the concentration of PTX, 6α-OHP, and 3′-OHP in plasma in the absence of resveratrol, respectively.

**Figure 5 molecules-28-01027-f005:**
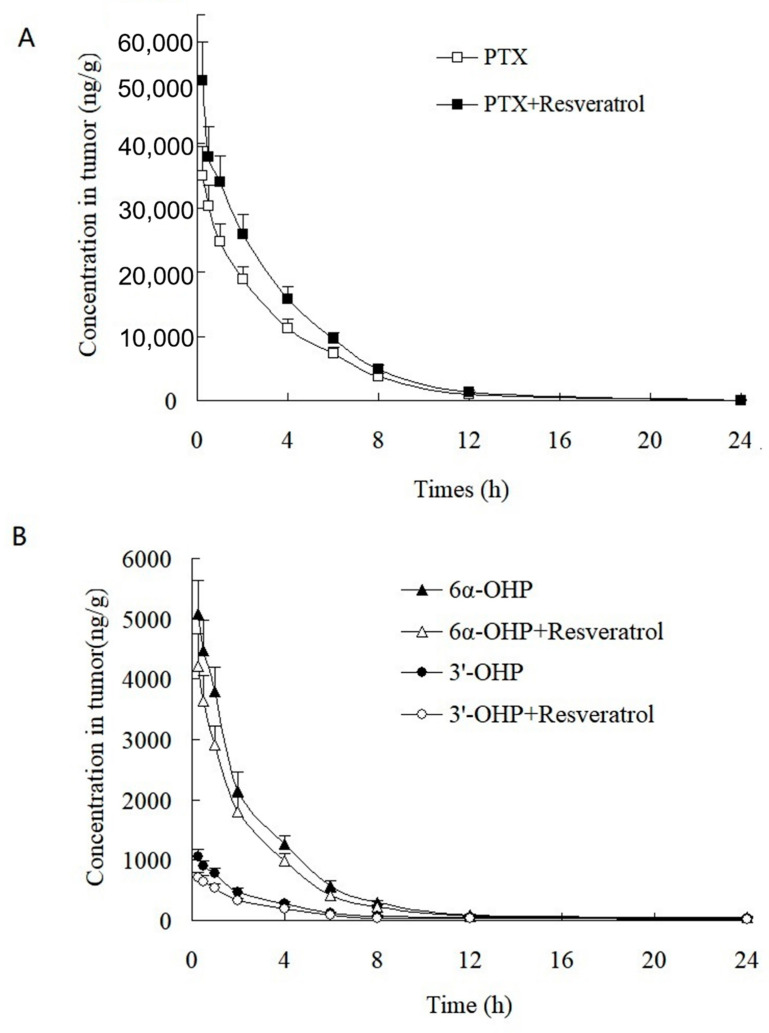
Tumor concentration–time profiles of (**A**) PTX, (**B**) 6α-OHP, and 3′-OHP in post xenograft nude mice intravenously administered with PTX solution at 5 mg/kg. ■, △, and ○: the concentration of PTX, 6α-OHP, and 3′-OHP in tumor in the presence of resveratrol, respectively; ☐, and ▲, and ●: the concentration of PTX, 6α-OHP, and 3′-OHP in tumor in the absence of resveratrol, respectively.

**Table 2 molecules-28-01027-t002:** Standard curves, correlation coefficients, linear ranges, and LLOQ.

Compound	Standard Curves	Correlation Coefficients (*r*)	Linear Range (ng/mL)	LLOQ (ng/mL)
Plasma				
PTX	y = 0.023x + 0.00356	0.9912	0.5~1000.0	0.5
6α-OHP	y = 0.0343x + 0.00824	0.9982	0.25~500.0	0.25
3′-OHP	y = 0.0109x + 0.00289	0.9943	0.25~500.0	0.25
Tumor tissue homogenate				
PTX	y = 0.0292x + 0.00721	0.9971	0.5~1000.0	0.5
6α-OHP	y = 0.0391x + 0.00752	0.9965	0.25~500.0	0.25
3′-OHP	y = 0.0402x + 0.00947	0.9951	0.25~500.0	0.25

**Table 3 molecules-28-01027-t003:** Precision and accuracy of PTX, 6α-OHP, and 3′-OHP in mouse plasma and in tumor tissue homogenate (over three validation days, with six replicates at different concentration levels per day).

Compound	Concentration (ng/mL)		RSD (%)		RE(%)
	Added	Found (Mean)	Intra-Day	Inter-Day	
**Plasma**					
PTX	1.0	1.11	7.9	6.4	11.0
	500.0	544.6	8.7	9.1	8.9
	800.0	853.6	7.1	5.8	6.7
6α-OHP	0.50	0.53	9.1	8.4	6.0
	250.0	272.1	5.7	5.1	8.8
	400.0	439.4	6.1	4.4	9.8
3′-OHP	0.50	0.46	8.9	10.2	−8.0
	250.0	239.5	5.3	6.6	−4.2
	400.0	387.1	4.9	5.4	−3.2
**Tumor tissue homogenate**					
PTX	1.0	0.93	10.2	11.1	−7.0
	500.0	487.1	7.4	6.8	−2.6
	800.0	848.2	8.1	5.4	6.0
6α-OHP	0.50	0.52	9.2	9.6	4.0
	250.0	271.9	7.6	8.1	8.8
	400.0	432.6	7.3	5.2	8.2
3′-OHP	0.50	0.48	9.4	7.2	−4.8
	250.0	269.4	7.6	8.9	7.8
	400.0	423.8	4.3	6.1	6.0

**Table 4 molecules-28-01027-t004:** Mean pharmacokinetic parameters of PTX in plasma after i.v. administration of PTX (5 mg/kg) to xenografted nude mice in the presence or absence of resveratrol.

Parameters	PTX	PTX + Resveratrol
AUC (ng·h/mL)	7506.8 ± 809.3	9799.2 ± 783.1 *
Cl_t_ (mL/h)	14.1 ± 1.6	10.2 ± 1.3 *
*k* (h^−1^)	0.27 ± 0.021	0.24 ± 0.031
*t*_1/2_ (h)	2.62 ± 0.23	2.90 ± 0.34

* *p <* 0.05, compared with no resveratrol.

**Table 5 molecules-28-01027-t005:** Mean pharmacokinetic parameters of 6α-OHP and 3′-OHP after i.v. administration of PTX (5 mg/kg) to xenografted nude mice in the presence or absence of resveratrol.

Parameter	6α-OHP		3′-OHP	
	Without Resveratrol	Resveratrol	Without Resveratrol	Resveratrol
AUC (ng·h/mL)	922.4 ± 103.2	663.8 ± 79.4 *	196.4 ± 23.6	126.7 ± 23.1 *
*t*_1/2_ (h)	1.77 ± 0.12	1.65 ± 0.14	3.03 ± 0.05	2.17 ± 0.26
*K* (h^−1^)	0.39 ± 0.031	0.42 ± 0.045	0.23 ± 0.04	0.32 ± 0.04

* *p <* 0.05, compared with no resveratrol.

## Data Availability

The data presented in this study are available on request from the corresponding author.

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
