# Peer review of "The Quantification of Paclitaxel and Its Two Major Metabolites in Biological Samples by HPLC-MS/MS and Its Application in a Pharmacokinetic and Tumor Distribution Study in Xenograft Nude Mouse"

_molecules, 2023, doi:10.3390/molecules28031027_

Round 1

Reviewer 1 Report

The study made in this work was well performed and planned, the manuscript is quite clear and well organized. However, some issues should be clarified and the English grammar should be revised, before publication

Some author is missing? Because in the list of authors says: “Q. Liu and5*…..”

Page 2: define AUC, although it is a quite known acronym.

Page 3: “LC-VP HPLC” written in this form, seems to refer to a type or mode of LC, but refers to the equipment series (VP Series) so, it is not relevant to include VP in the acronym.

Section 3.1: The authors made different sample pretreatment and they have manifested that the better method was the SPE one. However, they do not show any result. Is should be interesting to show to the scientific community that SPE performed better than protein precipitation, liquid-liquid extraction and so on. E.g., which solvents have tried for LLE? How they have made protein precipitation? The experiments and results (Figures, Tables..) should be exposed.

If the polar interferents elute with the dead volume, and the analytes are well retained, then it should be not necessary to clean the sample….The paragraph of Section 3.1. should be better explained.  

In the same Section, the authors say that the total analysis time (3 min) was much shorter than the ones reported by other authors. The authors should give appropriate citations from the literature. Also, a Table should be made comparing the different figures of merit (precision, accuracy, recovery, and so on….) of the proposed method with other methods reported in the literature.

“…formic acid ……could increase the mass response of analytes…”. Formic acid increase or not the signal? The mass spectra without formic acid should be inserted.

Table 1: r (should be R) is the correlation coefficient, R2, the correct parameter to measure linearity, is the “determination coefficient”.

Table 2: accuracy and RE (relative error) are the same? Clarify at the bottom of the Table. Include in the Text the used equations since it is not clear. If they are expressed as %, the name should be “percentage relative error”.

Section 3.2.4: How the matrix effects were calculated? Please, include the used equation.

Table 4: replace “no resveratrol” by “without resveratrol”

Report in the Conclusions, the numbers for the obtained figures of merit of the analytical procedure for each studied compound.

A similar study to the present manuscript was made by Posocco, Bianca et. al PLoS ONE, 13, 2018 : “A new high-performance liquid chromatography-tandem mass spectrometry method for the determination of paclitaxel and 6α-hydroxy-paclitaxel in human plasma: Development, validation and application in a clinical pharmacokinetic study”. (DOI: 10.1371/journal.pone.0193500). This work should be cited.

English grammar should be revised, e.g., in Section 2.6 “Plasma and tumor cells(?) were collected….”. Section 3.1: “To realize ..” should be replaced by, e.g., “To achieve…”. Section 3.2.3: “All results meet the FDA criteria for a bioanalytical….., which show that the present…..”

Reviewer 2 Report

Dear authors,

congratulations on a good article.
There are no major comments, only the conclusions can be improved. More can be found in the article, e.g. "The oral administration of resveratrol for five days significantly (p<0.05) increased the AUC of PTX by 30.1% and decreased the total plasma clearance (Clt ) of PTX by 27.7%"

King regards

Reviewer 3 Report

The article mentions the uncertain word "possible" many times, and hopes to use it carefully.

Round 2

Reviewer 1 Report

Some of the most important changes required by this Reviewer were not done. I repeat my previous comments bellow. The paper should not be published before those important changes.

1) Section 3.1: The authors made different sample pretreatment and they have manifested that the better method was the SPE one. However, they do not show any result. Is should be interesting to show to the scientific community that SPE performed better than protein precipitation, liquid-liquid extraction and so on. E.g., which solvents have tried for LLE? How they have made protein precipitation? The experiments and results (Figures, Tables..) should be exposed.

2) In the same Section, the authors say that the total analysis time (3 min) was much shorter than the ones reported by other authors. The authors should give appropriate citations from the literature. Also, a Table should be made comparing the different figures of merit (precision, accuracy, recovery, and so on….) of the proposed method with other methods reported in the literature.

3) “…formic acid ……could increase the mass response of analytes…”. Formic acid increase or not the signal? The mass spectra without formic acid should be inserted.
